# Bactericidal Chitosan Derivatives and Their Superabsorbent Blends with ĸ-Carrageenan

**DOI:** 10.3390/ijms25084534

**Published:** 2024-04-20

**Authors:** Kamila Lewicka, Anna Smola-Dmochowska, Natalia Śmigiel-Gac, Bożena Kaczmarczyk, Henryk Janeczek, Renata Barczyńska-Felusiak, Izabela Szymanek, Piotr Rychter, Piotr Dobrzyński

**Affiliations:** 1Faculty of Science and Technology, Jan Dlugosz University in Czestochowa, 13/15 Armii Krajowej Av., 42-200 Czestochowa, Poland; k.lewicka@ujd.edu.pl (K.L.); r.barczynska-felusiak@ujd.edu.pl (R.B.-F.); izabela.szymanek@doktorant.ujd.edu.pl (I.S.); p.rychter@ujd.edu.pl (P.R.); 2Centre of Polymer and Carbon Materials, Polish Academy of Sciences, 41-819 Zabrze, Poland; asmola@cmpw-pan.pl (A.S.-D.); bkaczmarczyk@cmpw-pan.pl (B.K.); hjaneczek@cmpw-pan.pl (H.J.)

**Keywords:** biodegradable polymers, antibacterial polymers, chitosan Schiff base, κ-carrageenan blend, grafted copolymer

## Abstract

The aim of this work is research dedicated to the search for new bactericidal systems for use in cosmetic formulations, dermocosmetics, or the production of wound dressings. Over the last two decades, chitosan, due to its special biological activity, has become a highly indispensable biopolymer with very wide application possibilities. Reports in the literature on the antibacterial effects of chitosan are very diverse, but our research has shown that they can be successfully improved through chemical modification. Therefore, in this study, results on the synthesis of new chitosan-based Schiff bases, dCsSB-SFD and dCsSB-PCA, are obtained using two aldehydes: sodium 4-formylbenzene-1,3-disulfonate (SFD) and 2-pyridine carboxaldehyde (PCA), respectively. Chitosan derivatives synthesized in this way demonstrate stronger antimicrobial activity. Carrying out the procedure of grafting chitosan with a caproyl chain allowed obtaining compatible blends of chitosan derivatives with κ-carrageenan, which are stable hydrogels with a high swelling coefficient. Furthermore, the covalently bounded poly(ε-caprolactone) (PCL) chain improved the solubility of obtained polymers in organic solvents. In this respect, the Schiff base-containing polymers obtained in this study, with special hydrogel and antimicrobial properties, are very promising materials for potential use as a controlled-release formulation of both hydrophilic and hydrophobic drugs in cosmetic products for skin health.

## 1. Introduction

The modern world faces serious threats related to infections caused by bacteria and other microorganisms [1]. The widespread and enormous increase in the use of antibiotics is the main cause of microbial resistance to antimicrobials. Especially during the COVID-19 pandemic, antibiotic treatment was used on a massive scale, practically without medical supervision. To effectively fight multidrug-resistant bacteria, it was necessary to look for stronger or more complex antibiotic formulas [2]. Polymers, especially those containing nitrogen-based groups, possess not only antibacterial properties but also are not toxic; indeed, they become a nutrient for plants and therefore are becoming increasingly important not only in medicine and health care but also in cosmetics, food, and also in textile industries or agriculture [3,4,5,6].

Chitosan (Cs) is one of the most representative and extensively investigated biodegradable and biocompatible polysaccharides obtained from biopolymer chitin synthesized from N-acetyl-D-glucosamine units. These units form covalent β-(1→4)-linkages. It is industrially produced by deacetylation of chitin by treatment with sodium hydroxide. As a representative of antibacterial agents [2,7], Cs has broad-spectrum antibacterial properties, but its antibacterial ability is insufficient and limited by many factors such as poor solubility in water and organic solvents, high viscosity of solutions, and low drug-loading capacity [8,9,10]. However, Cs contains a large number of amino and hydroxyl groups, which allow chemical modifications by the introduction of other functional groups into the molecular chain to improve its functionality and increase its utilization value [10,11]. Recently, the attention of researchers has been drawn toward the modification of chitosan to enhance its biological activities [11,12].

One of the possibilities of modifying chitosan is reactions of the free amino groups of chitosan with an active carbonyl compound, ketone, or aldehyde to form chitosan Schiff base polymers (CsSBs) [13,14,15], for example, Schiff bases formed by the reaction of chitosan with aldehydes such as benzaldehyde [16,17], salicylaldehyde [18,19], and heteroaryl aldehydes [20]. Using dialdehydes/diketones such as glutaraldehyde [21], glutaraldehyde–thiourea mixture [22], and dialdehyde cellulose [23] in the cross-linking technique, CsSB gels can be obtained. The chitosan Schiff base polymers are promising materials with antibacterial [24,25], antifungal [26,27], and anticancer [28,29] properties.

Another method of chitosan modification is graft polymerization which improves the native properties of chitosan [30]. Chitosan modification by grafting has been explored as a convenient method to prepare new biohybrid materials such as chitosan-graft poly(ε-caprolactone) [31], chitosan-graft polylactide [32], and chitosan-graft poly(ethylene glycol) [33]. Synthetic polyesters such as polylactide (PLA), poly(ε-caprolactone) (PCL), poly(ethylene glycol) (PEG), and their copolymers have been widely used in biomedical fields. The disadvantages of these biodegradable polymers are their hydrophobicity and the lack of functional groups in their molecular chains. Therefore, the grafting of polyesters onto Cs may yield novel amphiphilic copolymers consisting of the hydrophilic segments of cationic chitosan and the hydrophobic segments of polyester [34,35]. These hybrids are completely biodegradable and are expected to form a micelle structure in an aqueous environment with a cationic and hydrophilic outer shell and a hydrophobic inner core. Additionally, they have been proven to be effective delivery vehicles for poorly water-soluble drugs and can also generate a cationic surface to promote cell adhesion when used as tissue engineering scaffolds [36,37].

Next to chitosan, a widely used natural polysaccharide is carrageenan. Carrageenan (CG) is a sulfated linear polysaccharide obtained from certain species of red seaweeds. CG is non-toxic, has high biological activity, and has great potential as a film-forming material. In addition, it is widely used in the food industry, as well as in pharmaceutical, cosmetics, printing, and textile formulations [38]. Due to its mechanical properties, particularly its gelling capacity and structure similar to glycosaminoglycans present in the native extracellular matrix (ECM), CG has recently begun to be explored as a biomaterial for tissue engineering applications including drug or growth factor delivery systems, enzyme confinement, and cell encapsulation for in vivo delivery [39,40].

In many cosmetic and medical applications, a very valuable property of biocompatible materials is the ability to absorb large amounts of water and produce hydrogels [41,42]. Carrageenans can form hydrogels that can be used in many biomedical applications, e.g., in controlled drug release processes, tissue engineering, or wound dressing [43]. Chitosan and its derivatives can also form hydrogels, but this requires the creation of a network structure of the molecules of these compounds through physical [44] or chemical cross-linking [45,46]. Hydrogels obtained by preparing blends of chitosan with other polymers seem to be particularly interesting from the point of view of medical applications [47,48] among which the most outstanding are composites formed from positively charged chitosan and negatively charged carrageenan [49,50,51]. Physical hydrogels based on carrageenan/chitosan prepared by electrostatic complexation in aqueous solutions were reported by Papagiannopoulos et al. [52]. Aranilla et al. [53] described carboxymethyl *κ*-carrageenan and chitosan hydrogels obtained via radiation cross-linking. Obtained polymeric materials were a hemostatic agent and bleeding controller. Khalil et al. prepared carrageenan/chitosan antibacterial hydrogels loaded with silver and/or copper nanoparticles for wound dressing applications [51].

In this study, results on the synthesis of Cs-g-PCL copolymers and their compatible hydrogel blends with ĸ-CG are presented. The literature describes a variety of catalysts such as Sn(Oct)_2_, Ti(OBu)_4_, 4-dimethylamino pyridine, and enzymes, used for the synthesis of Cs-g-PCL copolymers in which hydroxyl and/or amino groups of Cs were used as initiating species [54,55]. However, it is important to maintain various specific functions of the aminosaccharide units, including biological activities and polycationic nature. Therefore, Duan et al. (2010) developed the one-step approach to synthesize amino-reserved Cs-g-PCL by grafting ε-CL onto the hydroxyl groups of Cs via ring-opening polymerization where the methanesulfonic acid (MSA) plays a dual solvent and catalyst role. As a result of the protective protonation of the Cs amino groups in an acidic medium, the grafting of ε-CL takes place mainly on the Cs hydroxyl groups and these amino group-reserved Cs-g-PCL copolymers would be able to generate a cationic surface to promote cell adhesion when used as tissue engineering scaffolds [35].

An important aspect of the presented study is the synthesis of new chitosan-based Schiff bases, dCsSB-SFD and dCsSB-PCA, which are obtained using two aldehydes: sodium 4-formylbenzene-1,3-disulfonate (SFD) and 2-pyridine carboxaldehyde (PCA), respectively. Although there are several reports in the literature dealing with the modification of chitosan using PCA [10,56,57,58], according to our knowledge SFD has not yet been used in the synthesis of CsSBs. Structurally, an attractive feature in the aldehyde chosen is the presence of sulfonate groups which can enhance the antibacterial effect and have antithrombotic properties.

In this paper, a series of chitosan derivatives were synthesized and antibacterial and antifungal studies of them have been conducted with a view to their potential in the cosmetic products. It has been confirmed that the polymers obtained can be used as a dressing material, preventing or treating a possible wound infection, making the skin healing process faster and smoother, as well as creating drug carriers and other active compounds in the form of microparticles, especially as an ingredient bactericidal creams and ointments used in dermatology or cosmetology.

## 2. Results

### 2.1. N-Deacetylation of Chitosan

The degree of deacetylation (DD) is the most important parameter affecting the properties of chitosan, as well as its antibacterial activity. It was confirmed that high chitosan DD (more than 95%) inhibited the growth of almost all types of tested bacteria at the lowest concentration, and the antibacterial activity is the highest [59].

The higher the degree of chitin deacetylation, the higher the reactivity of chitosan occurring through amino groups. There are several methods to determine the degree of deacetylation of chitosan by measuring its free amine groups. For chitosan and deacetylated chitosan (dCs), the degree of DD was determined from ^1^H-NMR spectra (Figure 1, Table 1). These values were calculated according to the equation (1) given in the literature [60], determined from the ratio of areas of the proton f (-CH(NH_2_)-) of deacetylated monomer related to the peak of the three protons of acetyl group j (CH_3_CO-). Molecular weight for Cs and dCs was determined based on inherent viscosity tests using Equation (2) [61]. Table 1 presents the results before and after the deacetylation process. Based on the results, it can be concluded that the method of deacetylation did not significantly decrease the molecular weight.

### 2.2. Chemical Structure of dCs-ε-CL and dCs-ε-CL(MSA)

The first stage of our research was to obtain a chitosan derivative containing long aliphatic side chains. This structure of the copolymer, following previous observations, should not only reduce the solubility of chitosan in water, as well as allowing blends with many synthetic polyesters, such as polycaprolactone, to be obtained, but also increase the antibacterial activity of chitosan due to the possibility of a stronger destructive effect on the bacterial cell wall [62]. These types of chains were introduced into the molecule through the grafting reaction of chitosan with ε-caprolactone. This process was carried out using two methods. In the first method, copolymerization was carried out in a DMSO solution, carrying out ROP of ε-caprolactone initiated by the zinc complex Zn[(acac)(LPhe)H_2_O] containing ligands composed of acetylacetonate groups and Schiff bases (LPhe) obtained during the condensation of phenylalanine with pyridine carboxaldehyde [63]. The choice of the initiator was determined by the practical lack of cytotoxicity and the confirmed high efficiency of this initiator during the polymerization of ROP lactide or lactide copolymerization, as well as its additionally confirmed high bactericidal activity [64]. Polymerization was carried out at different molar ratios of glucopyranoside repeating units of chitosan/ε-caprolactone, obtaining chitosan grafted through an amine group with a caproyl chain (dCs-ε-CL). In the second method used, the process was carried out in a methanesulfonic acid environment, modified by the method described earlier, to obtain copolymers dCs-ε-CL(MSA) containing caproyl chains grafted onto the hydroxyl groups of chitosan [65].

The chemical structures obtained copolymers of dCs-ε-CL and dCs-ε-CL(MSA) were verified by ^1^H-NMR and FTIR spectra. In the ^1^H-NMR spectrum, as illustrated in Figure 2a for dCs-ε-CL, the peaks at 2.8–3.0, 3.4–3.8 ppm were assigned to f, g, h, and i protons in Cs units according to the previous assignment of signals in the poor chitosan spectrum (Figure 1). The e, a, b, c, d signals were shown at the peaks of protons -(CO)OCH_2_-, (CO)CH_2_, (CO)CH_2_CH_2_-, (CO)CH_2_CH_2_CH_2_-, and (CO)CH_2_CH_2_CH_2_CH_2_- in caproyl units, respectively (Figure 2a). This spectrum also shows the second series of signals e′, a′, b′, c′, and d′, which were assigned to analogous protons of the appropriate groups of caproyl units terminating the grafted PCL chains.

In the case of the dCs-ε-CL(MSA) spectrum, in addition to the signals e, a, b, d, and c mentioned above, there were also very strong analogous signals E, A, C, and B + D (Figure 2b). These are signals characteristic of high-molecular-weight PCL [66]. They correspond to the protons of the caproyl units of the grafted PCL.

The copolymers obtained in the ROP reaction initiated with the zinc (II) complex, despite the five times stoichiometric excess of caprolactone used, ultimately contained a similar, quite small amount of attached caprolactone units. Based on the ratio of signal intensities i to (A + a)/2 (Figure 2a), the molar ratio of glucopyranoside chitosan units to caproyl units was calculated and amounted to approx. 1:40. At the same time, the presence of intense signals related to the protons of the end group units of the created caproyl chains indicates the short length of the PCL sequences. The calculated average length of the created caproyl blocks (from the intensity of the a and a′ signals) was only about five caproyl units. At the same time, among the signals related to the ring units, there are no visible signals f′ related to the presence of protons of –CH(NH_2_)- glucopyranoside ring groups, but there are also visible f signals (2.9 ppm) assigned to the signals of protons –CH(NH-CO(CH_2_)_5_OCO(CH_2_)_5_)_n_–, formed as a result of the growth of the caproyl chain on the amino groups. The above data show that during the grafting reaction initiated with the zinc (II) complex, a chitosan copolymer was obtained with short caproyl sequences grafted on the amino groups. The hydroxyl groups of the chitosan ring obtained via ROP initiated with the zinc (II) complex did not undergo the grafting reaction (Figure 2a).

The chitosan copolymer obtained in a reaction involving methanesulfonic acid has a completely different structure. In this case, we obtained copolymers with long caproyl blocks (signals: E, A, B, and C + D). In the spectrum of this compound there is also a strong f′ signal (3.05 ppm) associated with the presence of protons of the -CH(NH_2_)- groups, which indicates that the growth of caproyl blocks took place mainly on hydroxyl groups. The molar ratio of caproyl units to glucopyranoside chitosan units ranged from 10 to 20, depending on the starting stoichiometric excess of ε-caprolactone used. The presence of weak signals e, a, b, c, and d, located practically identically as in the previous spectrum, show that some of the amino groups were subject to a grafting reaction, which resulted in the formation of a relatively small amount of caproyl sequences (up to about 10% of the total caproyl units) grafted onto the amino groups.

The rate of the chitosan grafting process was also observed by the following changes in the FTIR spectrum, as illustrated in Figure 3. For the Cs, the band at 890 cm^−1^ corresponded to the C-O stretching vibrations of glycoside in the polysaccharide structure and the band at 1147 cm^−1^ was related to asymmetric C-O stretching vibrations of C-O-C. The bands at 1614 cm^−1^ and 1640 cm^−1^ were attributed to deformation vibrations C-H in amine groups. In the case of the dCs-ε-CL spectrum, the signals related to the amino groups become less intensive, while signals related to C=O stretching vibrations of amide (I) bands appeared. Peaks at 2862 cm^−1^ and 2945 cm^−1^ were related to C-H stretching vibrations of Cs which have been intensified in copolymer spectrum accumulated by C-H stretching vibrations of CL units. At 3445 cm^−1^, O-H and N-H stretching bands of chitosan overlapped. These signals occur in all the tested copolymers, which suggests that in the case of dCs-ε-CL(MSA) the growth of grafted PCL chains takes place only on part of the hydroxyl groups of chitosan units. The characteristic absorbance of the PCL ester bond appeared at 1725 cm^−1^. The copolymers of dCs-ε-CL and dCs-ε-CL(MSA) presented absorbance bands at 1725, 1640, and 1614 cm^−1^, which were assigned to the characteristic bands of ester in PCL, amide I band, and amino groups in Cs, respectively. The appearance of this signal in the spectra of dCs-ε-CL and dCs-ε-CL(MSA) confirms the grafting process of PCL onto Cs through its amino groups [67,68,69]. Due to the much higher signal intensity at 1725 cm^−1^ of the spectrum of the copolymer obtained during the grafting reaction with methanesulfonic acid, the higher content of caproyl units in this copolymer compared to the analogous one obtained with the zinc complex was also confirmed.

### 2.3. Chemical Structure of Schiff Bases dCsSB-PCA and dCsSB-SFD and Copolymers dCsSB-PCA-ε-CL and dCsSB-SFD-ε-CL

Fabrication of the antibacterial material was carried out by condensation reaction of deacetylated chitosan (dCs) with 2-pyridine carboxaldehyde (PCA) or sodium 4-formylbenzene-1,3-disulfonate aldehyde (SFD) via Schiff base formation. This reaction was carried out with a 20% stoichiometric excess of aldehyde (calculated concerning amino groups of chitosan). In this way, chitosan derivatives with significantly different properties were obtained. A Schiff base obtained from pyridine carboxaldehyde (CsSB-PCA), was slightly soluble in water and easily soluble in selected organic solvents. The chitosan derivative obtained by condensation reaction with the sodium salt of 4-formylbenzene-1,3-disulfonate (dCsSB-SFD) was practically insoluble in organic solvents.

The structure of the obtained chitosan-based Schiff bases was confirmed by ^1^H-NMR (Figure 4 and Figure 5). Comparing the obtained modified chitosan CsSB-PCA with Cs, the appearance of new signals was noticed. Signal 1 at 8.8 ppm corresponded to the imine protons (-N=CH-ArN)-, while those at 8.4; 8.1; 7.9, and 7.7 ppm correspond to the 6, 5, 3, and 4 protons on the pyridine ring residues, respectively. The appearance of signal 7 (4.75 ppm) associated with the proton of chitosan units near the imide group (-CH-(N=CH-ArN) was also noticed. However, the signal f (2.8 ppm) associated with the analogous proton near the amino group was still visible (-CH-NH_2_). The degree of substitution (DS) was calculated from the ratio between the integrated resonances of the hydrogen at carbon 1 in the imine groups and the sum of protons (f + 7) of the glucopyranoside ring. It was calculated that despite the large excess of aldehyde only about 45% of the amino groups reacted.

The ^1^H NMR spectrum of dCsSB-SFD was obtained in D_2_O (Figure 5) because the product was insoluble in DMSO. Also, in this case, signals analogous to the spectrum discussed earlier appeared too. Signal 1 (9.3 ppm) was assigned to protons –N=CH–Ar(SO_2_)_2_, signal 5 (3.8–4.0 ppm) to protons of the chitosan ring units -CH(N=CH-Ar(SO_2_)_2_)-. A quite strong characteristic signal of the protons of the chitosan units connected to the amino group, the signal f -CH(NH_2_), was also still visible. In this spectrum, there were also signals of the protons of the aromatic rings 4 and 2,3. The degree of amino group substitution was estimated at approximately 55%. In the next step, it was decided to obtain PCL-grafted chitosan containing the discussed imide side groups. For this purpose, the Schiff bases dCsSB-PCA and dCsSB-SFD reacted with ε-CL using zinc complex Zn[(acac)(LPhe)]H_2_O as an ROP initiator. The described method of chitosan copolymerization by ε-CL grafting with the presence of methanesulfonic acid could not be used in this case, because the acidic reaction environment caused rapid hydrolysis of the Schiff base. The reverse procedure was also difficult to carry out due to the insolubility of chitosan/caprolactone copolymers in solvents of the aldehydes.

The ^1^H NMR spectra of obtained copolymers are pictured in Figure 6 and Figure 7. In the spectrum of dCsSB-PCA-ε-CL (Figure 6), all signals appeared in the previously presented spectrum of the output Schiff base (Figure 4).

The protons described as the letters f, g, h, and j on the HNMR spectrum correspond to the chitosan moiety chain. The signals described in the numbers of this spectrum are associated with the Schiff base. The peak at 8.8 ppm corresponds to imine proton 7, and the signals at 8.1, 8.8, 7.9, and 8.3 ppm are associated with the aromatic protons on the pyridine ring residues. The protons described as the letters a, b, c, d, and e are assigned to caproyl units of PCL. However, unlike the previously presented dCs-εCL spectrum (Figure 2a), the signals assigned to the protons of the end groups of the caproyl block (a′, b′, c′, d′, and e′) were either very weak or invisible. This proves that the average length of the caproyl blocks in the dCsSB-PCA-εCL copolymer was much longer than in the dCs-εCL. There was also a clear f′’ signal (3.2 ppm) assigned to the protons of the -CH- groups of chitosan units located in the vicinity of the caproyl chains. However, a strong f signal associated with the protons of the -CH(NH_2_)- groups was still visible. The estimated degree of substitution of amino groups D in the obtained dCsSB-PCA-εCL, calculated according to the equation D = (f″ + 7)/(f + f″ + 7) × 100%, was approximately 70% in total (almost 60% substitution with Schiff bases and about 10% PCA chain). The composition of the obtained copolymer was estimated based on the intensity of signals f′, f, 7, 1, and a is 47 mol.% chitosan units, 37 mol.% Schiff base, 16 mol.% caproyl units.

The obtained ^1^H-NMR spectrum of the dCsSB-SFD-ε-CL with assigned signals is presented in Figure 7.

In this case, based on the intensity of the signals of the caproyl units a, b, c, d, and e, and the end groups of the caproyl blocks (signals b′, c′, d′ and e′), it can be seen that the average length of the block is similar to the length obtained during the grafting reaction of chitosan itself, initiated by the zinc (II) complex. There is an f′ signal in the spectrum, indicating the growth of caproyl blocks on amino groups. Signal 5 is also well visible coming from the protons of the -CH- group of chitosan linked to the imide group of the formed Schiff base. Moreover, signals characteristic of chitosan (g, h, j) and Schiff base (3, 2, 4, and 1) can be observed too. The composition of the obtained copolymer estimated based on the intensity of signals f′, 5, 1, and a was 40 mol% chitosan units, 20 mol% Schiff base, and 40 mol% caproyl units. The FTIR spectra of dCs, PCA, and the obtained Schiff base (dCsSB-PCA) and their copolymer with ε-CL (dCsSB-PCA-ε-CL) are shown in Figure 8a. The main changes that can be highlighted between the spectra of dCs and dCsSB-PCA are the presence of bands at 1571 cm^−1^ and 1432 cm^−1^ related to the stretching vibrations of the pyridine ring. The appearance of these bands is due to the introduction of a pyridine moiety into the chitosan backbone. In addition, the presence of a band at 776 cm^−1^ was also noticed, corresponding to the out-of-plane bending of C-H in the pyridine ring. These spectral data indicated that the chitosan modified by PCA had been successfully prepared. For the copolymer of dCsSB-PCA-ε-CL, the characteristic absorbance of the ester in PCL was demonstrated at 1731 cm^−1^, which implied the success of grafting PCL onto dCsSB-PCA [10,56,68]. Figure 8b shows the FTIR spectra of dCs, SFD, and the obtained Schiff base (dCsSB-SFD) and their copolymer with ε-CL (dCsSB-SFD-ε-CL). The characteristic bands at 1190 cm^−1^ and 1030 cm^−1^ are attributed to the S=O antisymmetric and S=O symmetric stretching bands. The small but significant characteristic peak at 960 cm^−1^ for the sulfonated samples may be due to the S–O–C stretching vibrations. For the dCsSB-SFD-ε-CL copolymer, the characteristic C=O band was observed to be slightly shifted to lower values at 1704 cm^−1^, which confirms the grafting of PCL onto dCsSB-SFD [35,69].

### 2.4. Hydrogel Blends with Carrageenan, Swelling Properties

Due to the possibility of using the developed copolymers in cosmetics and medicine, in many applications (moisturizing creams, dressings, drug-releasing systems), a particularly desirable property of such biodegradable and bactericidal materials is also the ability to create hydrogels with high water absorption above 1000%. For this purpose, the properties of carrageenan were used to create blends with this polysaccharide by dissolving it in water and mixing it with aqueous solutions of the produced chitosan derivatives. After the freeze-drying procedure and removing virtually all the water, a porous solid form was obtained and used for water absorption tests. In addition to polymer composites containing chitosan derivatives, these tests also included chitosan and carrageenan themselves for comparison.

Swelling properties are considered an important feature for the transportation of drugs from hydrogels because they have a great influence on the pore sizes of hydrogels [70]. The swelling ratio (SR) of prepared hydrogels as a function of time at room temperature and at 36 °C are presented in Figure 9. All tested materials belong to the group of high-swelling hydrogels [71]. Based on the obtained results, it was found that at room temperature all the obtained blends had a higher swelling ratio than both chitosan and κ-carrageenan. For all tested hydrogels, an increase in temperature also resulted in a visible increase in the degree of swelling.

The observed temperature-responsive swelling behavior is mainly related to the phenomenon of the association/dissociation of hydrogen bonding between the cations, amino groups, carbonyl groups, sulfates, and hydroxyl groups [72]. The highest SR is observed for dCs-ε-CL (MSA): CG 50:50 blend, which containing the highest amount of caproyl and the longest average length of caproyl blocks, much higher compared to the analogous blend dCs-ε-CL: CG 50:50 obtained in the ROP reaction initiated by the zinc (II) complex. For this reason, it seems that the presence of long, flexible chains of hydrophobic caproyl blocks that form hydrogen bonds between carboxyl groups and cations contained in both the carrageenan chain (sulfone groups) and the modified chitosan (amine, imine groups) have a decisive influence on the observed degree of swelling. At a slightly higher temperature, human skin temperature (36 °C), this difference decreased significantly. The degree of specification of the dCs-ε-CL: CG 50:50 blend is very similar to the blend.

In the case of unmodified chitosan and blends containing chitosan Schiff base, the dCsCB-SFD-ε-CL: CG 50:50 shows a higher degree of swelling. The reason might be that dCsCB-SFD-ε-CL: CG 50:50 is more hydrophilic than the dCsCB-PCA-ε-CL: CG 50:50. The highest SR for dCsCB-SFD-ε-CL: CG 50:50 was recorded after 24 h (SR = 1260%). At a temperature of 36 °C, after 3 h, the carrageenan and chitosan blends containing Schiff base gradually degrade and dissolve.

### 2.5. Thermal Properties

The thermal properties of ĸ-carrageenan, chitosan, and materials obtained based on them were investigated by the DSC method (Figure 10). Table 2 contains the glass transition and melting temperatures of the semi-crystalline phase. Because chitosan and ĸ-carrageenan can easily absorb water, the second run method is adopted to eliminate the influence of water. From the DSC curve of CG, it was possible to determine the glass transition temperature T_g_ = 88.2 °C and it is similar to what has been reported for ĸ-carrageenan powder [73]. The glass transition temperature for Cs was determined as T_g_ 173 °C. It is often difficult to determine the glass transition temperature of chitosan based on DSC curves, and there is a large variation between the values in the literature [74]. Nevertheless, the T_g_ value obtained here (173.0 °C) is comparable to the T_g_ reported by Martínez-Camacho et al. (170.9 °C) [75] and Pourjavadi et al. (188 °C) [50]. Two transitions can be observed in the DSC curve of ε-caprolactone grafted chitosan (dCs-ε-CL), which coincides with the glass transition temperature of the ε-caprolactone and chitosan blocks. A different effect is observed for chitosan grafted with ε-caprolactone in the presence of methanesulfonic acid dCs-ε-CL(MSA). Here, in the DSC studies, only the melting point for dCs-ε-CL(MSA) was observed at 50.6 °C (corresponding to the ε-CL chain fragment). This is because dCs-ε-CL(MSA) contains much longer ε-CL chains in its structure than dCs-ε-CL. The obtained Schiff bases only show the glass transition temperature, respectively, 141.4 °C for dCsSB-PCA and 102.4 °C for dCsSB-SFD, which indicated that the modification of chitosan with aldehyde effectively decreased the glass transition temperature of chitosan. After the grafting reaction with ε-caprolactone, two T_g_s are observed for dCsSB-PCA-ε-CL, respectively, −38 °C and 143 °C. The occurrence of two transitions is related to ε-caprolactone and chitosan blocks. In the case of dCsSB-SFD-ε-CL, two glass transition temperatures are also observed, but they are shifted to higher temperatures (20.4 °C and 155.4 °C). This is related to intermolecular interactions such as hydrogen bonding, which may limit the mobility of the polymer [76].

All prepared blends were compatible as confirmed by DSC. The dCs-ε-CL: CG 50:50 blend shows a slight increase in the glass transition temperature compared to dCs-ε-CL. Such an increase may be caused by intermolecular interactions like hydrogen bonding between the sulfonate groups of ĸ-carrageenan and the hydroxyl groups of chitosan. For dCs-ε-CL(MSA): CG 50:50 blend, the endothermic peak is shifted towards a lower temperature and the heat enthalpy has a lower value than for the dCs-ε-CL(MSA). The presence of the sulfonate group in the blends causes a lower melting point in DSC measurements. The same effect was described by T. Tanaka et al. for poly(vinyl alcohol)/ĸ-carrageenan blends [77]. It was found that the dCsCB-PCA-ε-CL: CG 50:50 blend shows a broad glass transition (T_g_ = 124.1 °C ΔT_g_ = 82.6 °C). In the case of dCsCB-SFD-ε-CL: CG 50:50, glass transition temperatures (−62.2 °C and 146.7 °C) are observed related to the presence of blocks derived from ε-CL and dCsCB-SFD. No glass transition temperature from CG was observed.

### 2.6. Antibacterial and Antifungal Evaluation

The aim of the research presented in this work was mainly to obtain a series of copolymers and hydrogels obtained with chitosan and carrageenan, and to select among them those with the highest antibacterial activity and presenting physico-chemical properties allowing their use as carriers of bioactive medicinal substances. Therefore, a much less time-consuming method than the MIC determination procedure was chosen to measure the antibacterial activity required to compare the obtained materials. For this purpose, studies were carried out on the impact of the tested copolymers on changes in the cell population of selected microorganisms, only at a concentration of 0.1 mg/ml, at which the activity of typical commercial chitosan was defined as “slight activity” [78]. All results presented below were obtained at this tested polymer concentration.

Figure 11 shows the antibacterial activity of the tested polymers against the *E. coli* strain. The greatest growth inhibition compared to the control was observed for samples containing Schiff bases. After 24 h, the growth of this strain decreased from 9.00 log_10_ CFU/ml to 5.68 log_10_ CFU/ml and 5.66 log_10_ CFU/ml for the dCsSB-PCA and dCsSB-SFD, respectively. E. coli growth was inhibited also by copolymers of Schiff bases with ε-CL and reached 6.23 log_10_ CFU/ml for dCsSB-PCA-ε-CL and 6.66 log_10_ CFU/ml for CsSB-SFD-ε-CL. Importantly, blends of these copolymers with CG show greater growth inhibition than the copolymers themselves and are 5.81 log_10_ CFU/ml dCsSB-PCA-ε-CL: CG 50:50 and 5.96 log_10_ CFU/ml CsSB-SFD-ε-CL: CG 50:50. Despite many reports in the literature proving the antibacterial activity of chitosan and carrageenan [59,79,80,81,82,83], only a slight decrease in the growth of the *E. coli* strain was observed at concentration 0.1 mg/ml. The antibacterial activity of this strain after 48 h is shown in Appendix A. The results were similar to those after 24 h.

The results regarding antibacterial activity against *P. aeruginosa* are presented in Figure 12. Similarly to the above, materials containing Schiff bases have the highest activity inhibiting the growth of this strain. Only these materials inhibit the growth of P. aeruginosa to a greater extent than the control. After 24 h, the growth of this strain decreased from 7.35 log_10_ CFU/ml to 5.56 log_10_ CFU/ml for dCsSB-PCA and 5.58 log_10_ CFU/ml for dCsSB-SFD. For chitosan, carrageenan, dCs-ε-CL, and dCs-ε-CL(MSA) copolymers and blends of these copolymers with carrageenan (dCsSB-PCA-ε-CL: CG 50:50 and CsSB-SFD-ε-CL: CG 50:50), a greater growth of this strain was observed compared to the control. In Appendix A the results after 48 h are presented, they also show no significant changes compared to 24 h.

Figure 13 presents the antibacterial activity against the *S. epidermidis* strain. The strongest growth inhibition was observed in the case of 24 h contact with dCsSB-PCA and dCsSB-SFD. The activity of this strain for these samples decreased from 8.84 log_10_ CFU/ml to 3.92 log_10_ CFU/ml and 4.11 log_10_ CFU/ml, respectively. A large decrease in activity was also observed for dCsSB-PCA-ε-CL to 4.21 log_10_ CFU/ml and dCsSB-SFD-ε-CL to 4.31 log_10_ CFU/ml. For this strain, both after 24 h and 48 h (Appendix A), all tested samples showed lower growth compared to the control.

In the case of antibacterial activity against the *S. aureus* strain shown in Figure 14, in addition to the inhibitory effect of substances containing Schiff bases, the effect of the copolymer of dCs-ε-CL(MSA) can be observed. Activity decreased from 7.80 log_10_ CFU/ml to 6.11 log_10_ CFU/ml The inhibitory effect of blends dCsSB-PCA-ε-CL: CG 50:50 and CsSB-SFD-ε-CL: CG 50:50 is also significant, with the activity of this strain decreasing to 5.51 log_10_ CFU/ml and 5.45 log_10_ CFU/ml, respectively. Similarly to the strains analyzed above, the bactericidal activity of *S. aureus* remains at a similar level after 48 h (Appendix A).

The antifungal activity against two selected strains *C. albicans* and *A. brasiliensis* is shown in Figure 14 and Figure 15. The growth inhibition of the *C. albicans* strain (Figure 15) was strongest in the culture with the addition of Schiff bases, copolymers of Schiff bases with ε-CL, and blends of these copolymers with CG. After 24 h, an increase in the strain was observed for chitosan (6.56 log_10_ CFU/ml), carrageenan (6.54 log_10_ CFU/ml), dCs-ε-CL (6.57 log_10_ CFU/ml), DCs-ε-CL(MSA) (6.54 log_10_ CFU/ml), and dCs-ε-CL: CG 50:50 (6.27 log_10_ CFU/ml) compared to the control (5.58 log_10_ CFU/ml). After 48 h (Appendix A), a decrease in antifungal activity was observed for all tested polymers. Interestingly, one of the greatest inhibitions of *C. albicans* growth was recorded for the dCS-ε-CL (4.48 log_10_ CFU/ml) with a control of 6.91 log_10_ CFU/ml.

Figure 16 shows the results of activity tests against the *A. brasiliensis* strain after 24 h. Only for the dCsSB-PCA and dCsSB-SFD, respectively, the same level (3.61 log_10_ CFU/ml) or minimal decrease (3.52 log_10_ CFU/ml) in activity was observed as for the control (3.61 log_10_ CFU/ml). For the remaining samples, the obtained values exceeded the values of the control sample. After 48 h (Appendix A), a further decrease in activity was observed for dCsSB-PCA (3.48 log_10_ CFU/ml) and dCsSB-SFD (3.45 log_10_ CFU/ml), as well as for dCsSB-PCA-ε-CL (3.45 log_10_ CFU/ml); dCsSB-SFD-ε-CL (3.45 log_10_ CFU/ml); dCsSB-PCA-ε-CL: CG 50:50 (3.36 log_10_ CFU/ml); and dCsSB-SFD-ε-CL: CG 50:50 (3.79 log_10_ CFU/ml) compared to the control (4.83 log_10_ CFU/ml).

In conclusion, regarding antibacterial and antifungal effects of the obtained polymers relative to unmodified Cs and CG, in all samples, a reduction in the growth of the tested microorganisms was observed. Both obtained Schiff bases (dCsSB-PCA; dCsSB-SFD) have a strong effect of inhibiting the growth of cells of the analyzed strains of bacteria and fungi. This effect is maintained in the case of copolymers of these Schiff bases (dCsSB-PCA-ε-CL; dCsSB-SFD-ε-CL) as well as blends of these copolymers with CG (dCsSB-PCA-ε-CL: CG 50:50; dCsSB-SFD-ε-CL: CG 50:50).

## 3. Discussion

Based on the literature, it can be concluded that reports regarding the antibacterial activity of chitosan vary greatly and are sometimes even contradictory. Our research has shown that the antimicrobial properties of chitosan may be successfully improved via its chemical modification. This is especially needed against gram-negative strains such as *E. coli* and *P. aeruginosa* because the antibacterial activity of chitosan against these strains is relatively weak.

Modification of chitosan by grafting this polysaccharide with caproyl chains caused changes in thermal properties, especially visible in copolymers containing long caproyl chains (dCs-ε-CL(MSA)). For these copolymers, the appearance of the melting temperature and the melting heat of the semi-crystalline phase of polycaprolactone was observed. This modification had little effect on the increase in antibacterial activity. Only dCs-ε-CL(MSA) copolymers showed greater activity compared to the initial unmodified chitosan against some bacterial strains; against *E. coli* (number of cells about 1 log_10_ CFU/ml lower for chitosan grafted with caprolactone after 24 h, similarly after 48 h concerning chitosan), and against *S. aureus* (difference of about 2 log_10_ CFU/ml after 24 h and about 0.5 log_10_CFU/ml after 48 h). dCs-ε-CL copolymers with short caproyl chains showed antibacterial activity very similar to unmodified chitosan. Copolymers with short caproyl blocks dCs-ε-CL showed increased activity compared to chitosan only against *C. albicans* after 48 h of exposure (cell number difference approximately 1.5 log_10_ CFU/ml).

A very effective and relatively simple way to increase the spectrum of antibacterial activity is to introduce imide groups by creating Schiff bases in the reaction of amino groups with selected aldehydes, SFD or PCA. The created chitosan derivatives (dCsSB-PCA and dCsSB-SFD) had only one glass transition temperature, significantly lower than the starting polysaccharide. Under the reaction conditions for the formation of an imine bond, only about half of the amino groups of chitosan were involved, despite the use of stoichiometric excesses of aldehydes. These modifications remarkably increased the antibacterial activity of chitosan. This was particularly visible against the *E. coli* strain where for both chitosan derivatives the number of cells in the test was lower by about 4.5 log_10_ CFU/mlcompared to chitosan; against *S. epidermis* for both derivatives there was a difference in the number of cells by about 3 log_10_ CFU/ml Interestingly, the dCsSB-SFD derivative containing sulfone groups was much more active against *S. aureus* (a difference of approximately 3 log_10_ CFU/ml). Chitosan derivatives synthesized in this way demonstrated stronger activity against all tested microorganisms. Similar chitosan derivatives, obtained in the grafting reaction of short caproyl chains on the remaining free amino groups of glucopyranoside chitosan units (dCsSB-PCA-ε-CL and dCsSB-SFD-ε-CL), presented practically the same activity against all tested microorganisms, despite the lack of side primary amino groups, in place of which amide groups of the caproyl chain were formed.

Perhaps the most interesting materials from the point of view of suitability for biomedical applications are blends obtained by mixing the same amounts of chitosan derivatives with carrageenan. All of them were compatible blends. The lack of glass transition temperature of carrageenan and the occurrence of hydrogen bonding interactions between the functional groups of modified chitosan and ĸ-carrageenan ensure the preparation of compatible blends. All of them also showed a high ability to absorb water, increasing with increasing temperature, clearly higher than the absorption capacity of chitosan or carrageenan. The highest swelling degree, reaching a value of 1800%, was demonstrated by the blend made of grafted dCs-ε-CL(MSA) chitosan containing long caproyl chains. The blend containing grafted dCS-ε-CL chitosan had similar absorbency, but only at skin temperature, i.e., 36 °C. The absorbency of the remaining blends oscillated between 1100% and 1600%.

To summarize, the swelling properties of the hydrogels strongly depend on composition. The analyzed hydrogels create a three-dimensional macromolecular network capable of absorbing water beyond their volume, which makes them attractive to use as drug carriers, for wound healing and in tissue engineering [84,85]. Importantly, the blends retained the antibacterial properties of the chitosan derivatives used for their formation. Despite the reduction in the concentration of imine bonds in the formed dCsCB-PCA-ε-CL: CG 50:50 and dCsCB-SFD-ε-CL: CG 50:50 blends, their activity against *E. coli*, *S. aureus*, *C. albicans*, and *A. brasillensis* remained practically as high as the activity of chitosan derivatives containing Schiff bases present in this composite. In other cases, the antibacterial activity of these blends was lower, but still clearly higher than the activity of chitosan.

The conducted research confirmed that the antimicrobial properties of chitosan can be improved by chemical modification of the Cs structure. The two reactive groups -NH_2_ and -OH offer vast opportunities for chemical modification. These groups allow the formation of several functional derivatives via reactions such as sulfonation and amination [86]. The materials described in this work are therefore promising for many medical and cosmetic applications. Of course, further research is required to determine the toxicity of the described copolymers, the rate of their degradation, and the possibility of using them in a specific application.

## 4. Materials and Methods

### 4.1. Reagents and Solutions

Low molecular weight chitosan (Cs) was purchased from Merck Life Science (Darmstadt, Germany), with a degree of deacetylation of 85%. Sodium 4-formylbenzene-1,3-disulfonate (SFD), 2-pyridinecarboxaldehyde (PCA), methanesulfonic acid (MSA) > 99.5%, ĸ-carrageenan, deuterium oxide (D_2_O), dimethyl sulfoxide (DMSO), trifluoroacetic acid (TFA) were purchased from Merck Life Science (Darmstadt, Germany). Sodium hydroxide (NaOH), acetic acid, sodium acetate, potassium dihydrogen phosphate (KH_2_PO_4_), and methanol were received from Avantor (Gliwice, Poland). All these chemicals were used as received. ε-caprolactone (ε-CL) was purchased from Acros Organics (Geel, Belgium) and purified using reduced-pressure vacuum distillation. Zinc complex Zn[(acac)(LPhe)H_2_O was prepared according to a previously published method [63].

### 4.2. The Strains and Substrates for Culture

The following strains were selected for testing: Gram-negative: *Escherichia coli* NCTC 12923/ATCC 8739; *Pseudomonas aeruginosa* NCTC 12924/ ATCC 9027; Gram-positive: *Staphylococcus aureus* NCTC 10788/ATCC 6538; *Staphylococcus epidermidis* NCTC 13360/ATCC 1222; Fungi: *Aspergillus brasiliensis* NCPF 2275/ATC C 16404; Yeast: *Candida albicans* NCPF 3179/ATCC 10231. The substrates were used for culture: E. coli—MacConkey Agar; *P. aeruginosa*—Cetrimide Agar; *S. aureus* and *S. epidermidis*—Chapman-Mannitol Salt Agar; *C. albicans* and *A. brasiliensis*—Sabouraud Dextrose Agar. All strains and subs were obtained from Biomaxima S.A., Centrum Mikrobiologii Biocorp, Poland. The tests were carried out for a polymer concentration in the medium of 0.1 mg/ml.

### 4.3. N-Deacetylation of Chitosan

*N*-Deacetylation of chitosan was carried out in a 250 mL glass reactor, equipped with a magnetic stirrer, a reflux condenser, and an argon gas supplier, placed in a controlled temperature thermal bath. A 45% NaOH aqueous solution was placed in the reactor and heated to a temperature of 90 °C. After that, 5 g of chitosan was added to the reactor and the reaction mixture was stirred at 90 °C for 3 h. Next, the mixture was cooled and the solid was filtered off, washed with distilled water until pH 7, and dried in an oven for 24 h at 40 °C.

### 4.4. Synthesis of the dCs-ε-CL Copolymer

dCs-ε-CL was synthesized by grafting ɛ-CL monomers onto functional groups of chitosan via ring-opening polymerization (Figure 1). Approximately 25 mL of anhydrous DMSO was placed in the 250 mL glass reactor and heated in an oil bath at 90 °C in an argon atmosphere with constant stirring. Next, 6 mL (0.054 mol) ε-CL was dissolved in DMSO, and with continuous stirring 3 g deacetylated of chitosan (dCs) and 0.027 g Zn[(acac)(LPhe)H_2_O] as the catalyst (molar ratio catalyst/ε-caprolactone as 1:1000). The reaction was carried out at 90 °C for 96 h. After the selected reaction time we removed the unreacted CL, free polycaprolactone (PCL), and the most used catalyst by extraction with chloroform. The remaining insoluble gel fraction was then dried in a vacuum, at 40 °C. The dry product after grinding was washed with water to remove eventual residues of ungrafted chitosan. Obtained dCs-ε-CL was left to dry in vacuum drying for constant mass.

### 4.5. Synthesis of the dCs-ε-CL (MSA) Copolymer

dCs-ε-CL(MSA) was synthesized by grafting ɛ-CL monomers via ring-opening polymerization (Figure 2) according to the modified procedure described elsewhere [35,87]. First 1 g of dCS (5.9 mmol glucosamine GA units), and 15 mL MeSO_3_H were charged into a 250 mL glass reactor and stirred for 30 min at 45 °C to allow dCS to dissolve, followed by the injection of 5.93 g ε-CL monomer (52 mmol, 8 equiv.). After 5 h, the resulting mixture was added to an ice-cold buffer solution containing 144 mL of KH_2_PO_4_ (0.2 M) and 23 mL of NaOH (10 M) and stirred for 30 min (the alkalinity of the solution is required to fully neutralize the MeSO_3_H). The mixture was kept at room temperature for 24 h to precipitate the solid phase out. After discarding the liquid phase, the solid product underwent precipitation to eliminate any leftover salts, followed by a wash with distilled water. The residual product, containing dCs-PCL copolymer and PCL homopolymer, was vacuum-dried overnight at 50 °C. Acetone was added to the solid powder and stirred for 4 h to separate the copolymer from the homopolymer. Precipitation was used to separate the homopolymer (soluble phase) from the copolymer (swelled insoluble phase).

### 4.6. Synthesis of the Schiff Base—dCsSB-PCA

As shown in Figure 3, in a 250 mL glass reactor 1.69 g of dCS (10 mmol GA units) was added to the 2% acetic acid (80 mL, 1%, *v*/*v*) and dissolved under stirring to form a transparent solution. An amount of 2.85 mL PCA (30 mmol GA units, 3 equiv.) was then added and the resulting mixture was stirred for 72 h at room temperature. During the addition of the aldehyde, the clear solution in the reactor becomes yellow. The solution was precipitated into excess acetone and the precipitant was filtrated and then vacuum-dried to a constant weight. The resulting product was milled in a cryogenic mill to give a light yellow powder.

### 4.7. Synthesis of the Schiff Base—dCsSB-SFD

The Schiff base (Figure 4) was synthesized using a previously reported method, with some modifications [17]. In a 250 mL glass reactor, 2.6 g deacetylated chitosan (dCs) was taken and dissolved in 80 mL 2% acetic acid. The mixture was stirred using a magnetic stirrer for 3 h at room temperature. To this, sodium 4-formylbenzene-1,3-disulfonate (SFD) (6 g) dissolved in 20 mL distilled water was added dropwise and the mixture was stirred continuously for 72 h at room temperature. During the addition of the aldehyde solution, the clear solution in the reactor becomes yellow. The mixture precipitated with cold methanol and then vacuum-dried to a constant weight. The resulting product was milled in a cryogenic mill to give a light yellow powder.

### 4.8. Synthesis of the dCsSB-PCA-ε-CL and dCsSB-SFD-ε-CL

The previously prepared Schiff bases (dCsSB-PCA and dCsSB-SFD) were used in the reaction with ε-caprolactone in the presence of Zn[(acac)(LPhe)H_2_O] as a catalyst of reaction. Approximately 25 mL of anhydrous DMSO was placed in the 250 mL glass reactor equipped with a magnetic stirrer and an argon gas supplier and heated to a temperature of 90 °C. Next, 6 mL (0.054 mol) ε-CL was dissolved in DMSO, and with continuous stirring added 3g dCsSB-PCA or dCsSB-SFD and 0.027 g Zn[(acac)(LPhe)H_2_O]. The reaction was carried out at 90 °C for 96 h. The obtained product was washed with chloroform to remove the unreacted CL, free polycaprolactone (PCL), and the most used catalyst. Figure 5 and Figure 6 present the reaction course of the obtained copolymers.

### 4.9. Preparation of the Cs and CG Blends

The aliquots of modified chitosan solutions (10 wt. % in 2% acetic acid) were slowly added to the carrageen solution (10 wt. % also in 2% acetic acid) to obtain blends in a 50:50 ratio. The final blends’ films were prepared by pouring these mixtures into Teflon dishes, drying initially for 48 h in air and then under vacuum at 40 °C to a constant mass so that the remaining solvent evaporates.

### 4.10. Characterization Methods

#### 4.10.1. Nuclear Magnetic Resonance (^1^H-NMR) Spectroscopy

The degree of chitosan deacetylation and structure of chitosan and its derivatives were determined with ^1^H NMR spectroscopy. The ^1^H-NMR spectra were recorded at 600 MHz Bruker Avance II Ultrashield Plus Spectrometer. ^1^H-NMR spectra were obtained with 64 scans, 11 µs pulse width, and 2.65 s acquisition time. DMSO/TFA or D_2_O were used as a deuterated solvent.

#### 4.10.2. Determination of Viscosity and Molecular Weight

The intrinsic viscosity [ⴄ] of chitosan samples was measured with the automatic Ubbelohde capillary viscometer in 0.2 M CH_3_COOH and 0.2 M C_2_H_3_NaO_2_ aqueous solution at 30 °C. The molecular weight of samples was calculated based on the appointed intrinsic viscosity [ⴄ] and Mark–Houwink equation.
(1)$$\left[ⴄ\right]=k{\left[Mv\right]}^{\alpha }$$
where constants: *Mν*—viscosity average molecular weight, constant: k = 1.4 × 10^−4^, α = 0.83 [61].

#### 4.10.3. Thermal Properties

By differential scanning calorimetry (DSC, DuPont 1090B apparatus calibrated with gallium and indium), thermal properties such as glass transition temperatures and heats of melting and crystallization of obtained copolymers were examined. The glass transition temperature was determined with a heating and cooling rate of 20 °C/min in the range between −100 and 220 °C, according to the ASTM E 1356-08 standard [88].

#### 4.10.4. Fourier Transform Infrared (FTIR) Spectroscopy

The FTIR spectra were recorded by an FTIR Nexus Nicolet spectrophotometer (Madison, WI, USA). Samples were analyzed in the form of KBr discs in the range of 4000–400 cm^−1^ and at a resolution of 1 cm^−1^ for 32 accumulated scans. The absorption spectra in the infrared region of the liquid samples (2-pyridine carboxaldehyde) were obtained using the same FTIR spectrophotometer, but it was equipped with the ATR accessory and zinc selenide crystal. The ATR crystal was carefully cleaned with acetone and dried using inert gas after each experiment to ensure the purest possible sample spectra.

#### 4.10.5. Swelling Ratio

The swelling ratio (SR) of the prepared hydrogels in phosphate-buffered saline solution (PBS) (pH 7.4) at room temperature and 36 °C was calculated according to equation [89]:(2)$$SR=\frac{Ww-Wd}{Wd}*100\%$$
where: *Ww* is the weight of the wet sample; *Wd* is the weight of the dry samples.

#### 4.10.6. Antibacterial and Antifungal Evaluation

The assessment of the antibacterial and antifungal activity and inhibitory concentrations were estimated using a microtiter broth dilution method, as recommended by the Clinical and Laboratory Standards Institute [90]. Each polymer sample was tested at a concentration of 0.1 mg/ml. Test tubes without tested compounds were used as the positive growth control. A diluted bacterial suspension was added to each test tube to yield a final concentration of 5 × 10^5^/5 × 10^6^ colony forming units (CFU)/ml, as confirmed by viable cell count (determined by turbidimetric method). The negative growth control was bacterial inoculum. The plates were incubated at 37 °C for 24 and 48 h. The contents of the test tube showing no visible growth were plated on selective substrates, and the number of colonies was counted after overnight incubation at 37 °C. For each strain, at least three independent determinations were performed, and the modal value was taken.

## 5. Conclusions

In the conducted research, biocompatible polymers with embedded Schiff bases as the active antimicrobial agent were successfully synthesized. Covalently bonded PCL improved the solubility of obtained polymers in organic solvents, which is very required in cosmetic products. More hydrophobic carriers of active agents in cosmetics facilitate their stability and enhance their skin availability. Polymers with embedded Schiff bases demonstrated much higher antimicrobial activity when compared to plain chitosan. Blends with nontoxic carrageenan enhanced the swelling properties of chitosan-based polymers and demonstrated antimicrobial activity lower than Schiff base polymers and higher than plain chitosan. In this respect, both polymers groups obtained within this study containing Schiff bases and their blends hydrogel-like properties are very promising materials for potential use as a controlled-release formulation of both hydrophilic and hydrophobic drugs in cosmetic products for skin health.

## Data Availability

The data presented in this study are available on request from the corresponding author.

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
