# Peer review of "Bactericidal Chitosan Derivatives and Their Superabsorbent Blends with ĸ-Carrageenan"

_ijms, 2024, doi:10.3390/ijms25084534_

Round 1
Reviewer 1 Report
Comments and Suggestions for Authors
The article “Bactericidal chitosan derivatives and their superabsorbent blends with ĸ-carrageenan” has a high applied value. It is well structured and generally makes a pleasant impression. The authors presented the NMR and FTIR spectroscopy results very well. However, there are some notes:
1. In section “2.6. Antibacterial and antifungal evaluation”, it would be better to present the MIC and MBC, values, as is customary in this type of research, rather than studying the effect of only one concentration of target polymers on microorganisms.
2. Section “3. Discussion" is written too laconically and does not contain a single reference. It should be expanded by comparing different types of chemical modification of chitosan and their effect on the antibacterial and antifungal properties of the resulting derivatives. As presented, this section looks more like a Conclusion.
3. In section “4.1. Reagents and solutions”, it is necessary to indicate the molecular weight of the chitosan that the authors used, because the molecular weight of chitosan or its derivatives is one of the factors that determine its antibacterial and antifungal properties. If the Merck Life Science does not indicate the molecular weight range of chitosan, it can be measured, for example, by determining the viscosity of its solution.
Technical Notes:
Line 44: in the word N-acetyl-β-D-glucosamine, the letter N should be italicized.
In the text of the article, the names of all strains (E. coli, P. aeruginosa, S. epidermidis, S. aureus, C. albicans, A. brasilliensis) should be italicized.
Author Response
The article “Bactericidal chitosan derivatives and their superabsorbent blends with ĸ-carrageenan” has a high applied value. It is well structured and generally makes a pleasant impression. The authors presented the NMR and FTIR spectroscopy results very well. However, there are some notes:
Thank you for your honest assessment of our work. In accordance with the reviewer's comments and instructions, with which we fully agree, we have introduced the suggested changes and additions to the manuscript.
In section “2.6. Antibacterial and antifungal evaluation”, it would be better to present the MIC and MBC, values, as is customary in this type of research, rather than studying the effect of only one concentration of target polymers on microorganisms.
We agree with the reviewer that the best way to express the antibacterial activity of a substance is to determine the MIC or MBC value. However, in the works published so far describing the activity of chitosan, other indicators were also used, the determination of which is less time-consuming and cheaper (inhibition zone, log CFU and others). This certainly makes it difficult to compare the results presented in many sources, but the determined MIC values often differ significantly, most likely because the tested activity of chitosan or its derivatives depends on the molecular weight, the origin of the polysaccharide, and conditions of the measurement (temperature, pH). This also brings complications in the comparison of measured activities.
The aim of the research presented in this work was mainly to obtain a series of copolymers and hydrogels obtained with chitosan and carrageenan, and to select among them those with the highest antibacterial activity and presenting physico-chemical properties allowing their use as carriers of bioactive medicinal substances. Therefore, a much less time-consuming method than the MIC determination procedure was chosen to measure the antibacterial activity required to compare the obtained materials. For this purpose, studies were carried out on the impact of the tested copolymers on changes in the cell population of selected microorganisms, only at a concentration of 0.1 mg/ml, at which the activity of typical commercial chitosan was defined as "slight activity" (Scientifc Reports | (2022) 12:8084 | https:/ /doi.org/10.1038/s41598-022-12150-3). Such a study made it possible to directly examine the relative impact of chitosan modifications on the increase in the antibacterial activity of such a copolymer or blend compared to the initial chitosan. Only in further application tests, conducted with the use of two or three selected hydrogels with optimal properties, will it be necessary to determine the MIC and MBC values for these materials against a number of selected bacteria.
The following explanation has been incorporated into the text of the manuscript (chapter: Antibacterial and antifungal evaluation; lines 425-434).
- Section “3. Discussion" is written too laconically and does not contain a single reference. It should be expanded by comparing different types of chemical modification of chitosan and their effect on the antibacterial and antifungal properties of the resulting derivatives. As presented, this section looks more like a Conclusion.
We have expanded the Discussion section as suggested by the reviewer.
Since most of the discussion of the obtained results was carried out in the earlier part in parallel with the description of the research results, fragments of the text were moved to the Discussion part. We have also supplemented the Discussion section with a broader description of the impact of individual modifications on the final properties of polymers and introducing extended conclusions from the conducted research.
- In section “4.1. Reagents and solutions”, it is necessary to indicate the molecular weight of the chitosan that the authors used, because the molecular weight of chitosan or its derivatives is one of the factors that determine its antibacterial and antifungal properties. If the Merck Life Science does not indicate the molecular weight range of chitosan, it can be measured, for example, by determining the viscosity of its solution.
We characterized the purchased chitosan by determining the intrinsic viscosity of its acidic aqueous solution. In viscosity measurements, we used an automatic Ubbelohde viscometer for polymer solutions. The measurement results are included in the section describing the course of deacetylation of the purchased chitosan, and the description is in the measurement methods. The average viscosity molecular weight Mv determined by us was approximately 500 kDa.
Technical Notes:
Line 44: in the word N-acetyl-β-D-glucosamine, the letter N should be italicized.
In the text of the article, the names of all strains (E. coli, P. aeruginosa, S. epidermidis, S. aureus, C. albicans, A. brasilliensis) should be italicized.
We have made these corrections.
Reviewer 2 Report
Comments and Suggestions for Authors
I remarked an interesting study about obtaining of new bactericidal systems with potential application as a controlled-release formulation of hydrophilic or hydrophobic drugs in skin health cosmetics.
The paper is very well organized, certainly consistent, with valuable results, but the writing should be improved. I have several comments that should be considered.
- Explain PCL abbreviation at first mention. Please, see Line 24.
- Escherichia coli, Pseudomonas aeruginosa, Staphylococcus aureus, Staphylococcus epidermidis, Aspergillus brasiliensis, ….. Please, write all of them in Italic format.
- On Line 44 there seems to be an error. “(1→4)”.
- In Chapter 4. Materials and methods, the authors used a code of notation for all the samples. If it is possible, for easier visibility, it would be useful to have a table showing the composition for each sample.
- In Chapter 4.10.4. Fourier Transform Infrared (FTIR) Spectroscopy, authors specified that samples were analyzed in the range of 4000–400cm−1, but other experimental details are missing! The authors should add information about the number of scans, and the spectral resolution.
I believe that the manuscript could be accepted for publication after a minor revision.
Comments on the Quality of English LanguageI remarked an interesting study about obtaining of new bactericidal systems with potential application as a controlled-release formulation of hydrophilic or hydrophobic drugs in skin health cosmetics.
The paper is very well organized, certainly consistent, with valuable results, but the writing should be improved. I have several comments that should be considered.
- Explain PCL abbreviation at first mention. Please, see Line 24.
- Escherichia coli, Pseudomonas aeruginosa, Staphylococcus aureus, Staphylococcus epidermidis, Aspergillus brasiliensis, ….. Please, write all of them in Italic format.
- On Line 44 there seems to be an error. “(1→4)”.
- In Chapter 4. Materials and methods, the authors used a code of notation for all the samples. If it is possible, for easier visibility, it would be useful to have a table showing the composition for each sample.
- In Chapter 4.10.4. Fourier Transform Infrared (FTIR) Spectroscopy, authors specified that samples were analyzed in the range of 4000–400cm−1, but other experimental details are missing! The authors should add information about the number of scans, and the spectral resolution.
I believe that the manuscript could be accepted for publication after a minor revision.
Author Response
I remarked an interesting study about obtaining of new bactericidal systems with potential application as a controlled-release formulation of hydrophilic or hydrophobic drugs in skin health cosmetics.
The paper is very well organized, certainly consistent, with valuable results, but the writing should be improved. I have several comments that should be considered.
Thank you very much for your positive opinion of our work. We have made changes to the manuscript in accordance with the reviewer's comments.
- Explain PCL abbreviation at first mention. Please, see Line 24.
We have entered the appropriate text
- Escherichia coli, Pseudomonas aeruginosa, Staphylococcus aureus, Staphylococcus epidermidis, Aspergillus brasiliensis, ….. Please, write all of them in Italic format.
- On Line 44 there seems to be an error. “(1→4)”.
Corrected in the text of the manuscript;
- In Chapter 4. Materials and methods, the authors used a code of notation for all the samples. If it is possible, for easier visibility, it would be useful to have a table showing the composition for each sample.
The table 3 has been introduced explaining the abbreviations used
- In Chapter 4.10.4. Fourier Transform Infrared (FTIR) Spectroscopy, authors specified that samples were analyzed in the range of 4000–400cm−1, but other experimental details are missing! The authors should add information about the number of scans, and the spectral resolution.
This data has been entered into the FTIR description text (line 711).
Round 2
Reviewer 1 Report
Comments and Suggestions for Authors
The authors took into account all my comments. The article can be published as presented.